# CoCas9 is a compact nuclease from the human microbiome for efficient and precise genome editing

Eleonora Pedrazzoli [1,6], Michele Demozzi[1,6], Elisabetta Visentin [1,6], Matteo Ciciani [1], Ilaria Bonuzzi [1], Laura Pezzè [2], Lorenzo Lucchetta[1], Giulia Maule [1], Simone Amistadi [1,3], Federica Esposito [4], Mariangela Lupo[4], Annarita Miccio [3], Alberto Auricchio [4,5], Antonio Casini [2], Nicola Segata [1,7] ✉ & Anna Cereseto [1,7] ✉

The expansion of the CRISPR-Cas toolbox is highly needed to accelerate the development of therapies for genetic diseases. Here, through the interrogation of a massively expanded repository of metagenome-assembled genomes, mostly from human microbiomes, we uncover a large variety ($n = 17,173$) of type II CRISPR-Cas loci. Among these we identify CoCas9, a strongly active and high-fidelity nuclease with reduced molecular size (1004 amino acids) isolated from an uncultivated *Collinsella* species. CoCas9 is efficiently co-delivered with its sgRNA through adeno associated viral (AAV) vectors, obtaining efficient in vivo editing in the mouse retina. With this study we uncover a collection of previously uncharacterized Cas9 nucleases, including CoCas9, which enriches the genome editing toolbox.

CRISPR tools greatly accelerated the development of curative therapies for genetic diseases. Yet, further development in the clinic strongly depends on the diversification of tools responding to the challenges of genome editing[1]. One of the most critical aspects is delivery, owing to the large molecular size of Cas nucleases and derived fusion products, including base editors[2]. The most efficient and widely used method for in vivo delivery are AAV vectors, which however are hardly compatible with CRISPR technologies due to size constraints (cargo smaller than 4.7 kbp)[2,3]. Indeed, the most commonly used Cas9 from *Streptococcus pyogenes* (SpCas9[4]) together with its single guide RNA (sgRNA) and all the necessary regulatory regions cannot be accommodated in all-in-one AAVs. Only few other Cas9s have been used with AAVs, mainly Cas9s from *Staphylococcus aureus* (SaCas9[5]) and *Neisseria meningitidis* (Nme2Cas9[6]), which however are less efficient nucleases than SpCas9. Overall, the diversity of CRISPR-Cas systems present in prokaryotic organisms has not been exploited yet.

Whole microbiome sequencing via metagenomics[7], followed by reconstruction of metagenome-assembled genomes (MAGs), has identified a huge variety of uncharacterized prokaryotic species[8–10], many of which encode phylogenetically diverse Cas9 orthologs[11,12]. Most quality-controlled MAGs are of high enough quality to enable full sequence characterization of whole Cas9 loci, and such uncovered diversity holds the potential for the discovery of nucleases with desirable properties to address the complexity of therapeutic applications.

Given the current limitations imposed by the molecular size of SpCas9, here we search for undescribed small Cas9 variants in a large database constituted by >154,000 microbial genomes reconstructed from >9400 human-associated metagenomes, including over 3000 previously uncharacterized bacterial species[8]. We identify and characterize CoCas9, a nuclease from a poorly characterized *Collinsella* species with compelling editing features in terms of editing efficiency

[1]Department of Computational, Cellular and Integrative Biology (CIBIO), University of Trento, 38123 Trento, Italy. [2]Alia Therapeutics, Trento, Italy. [3]Université de Paris, Imagine Institute, Laboratory of chromatin and gene regulation during development, INSERM, UMR 1163 Paris, France. [4]Telethon Institute of Genetics and Medicine (TIGEM), 80078 Pozzuoli (NA), Italy. [5]Medical Genetics, Department of Advanced Biomedical Sciences, University of Naples "Federico II", 80131 Naples, Italy. [6]These authors contributed equally: Eleonora Pedrazzoli, Michele Demozzi, Elisabetta Visentin. [7]These authors jointly supervised this work: Nicola Segata, Anna Cereseto. ✉e-mail: nicola.segata@unitn.it; anna.cereseto@unitn.it

and precision that can be packaged all-in-one with its sgRNA in AAV vectors for in vivo delivery.

## Results

### Identification of small Cas9 orthologs from human microbiome metagenomic data

We generated a computational pipeline for the identification of CRISPR-Cas9 systems. From this preliminary analysis we identified 33,978 *cas9* genes in the entire dataset, among which 17,173 were found associated with a complete locus composed of *cas1*, *cas2* and *cas9* genes and an adjacent CRISPR array. By integrating our dataset with 2125 Cas9 orthologs from public repositories, we generated a phylogenetic tree (see Methods), showing that variants identified from metagenomic data greatly expand the diversity of Cas9 proteins present in public databases (Fig. 1). The majority of Cas9 orthologs belong to 15 distinct bacterial phyla, with a pronounced prevalence observed in Bacillota, Pseudomonadota, Bacteroidota and Actinomycetota. Moreover, genera encoding a wide variety of Cas9 orthologs include *Streptococcus*, *Staphylococcus* and *Campylobacter*, from which the well-studied variants SpCas9, SaCas9 and CjCas9[13] were derived. Consistently with previous reports, short Cas9 orthologs belong to specific clades of subtypes II-A and II-C[11] (Supplementary Fig. 1).

We restricted our analysis to short Cas9 orthologs, ranging from 950 to 1100 amino acids, and selected a total of 436 proteins clustered at 95% sequence identity. Then, we identified a group of subtype II-C variants (*n* = 11) encoded in complete loci containing a large CRISPR array (> 5 spacers) and a readily identifiable tracrRNA (see Methods). After defining their PAM requirements using an in vitro cleavage assay (see Methods and Supplementary Fig. 2a), we tested their editing activity through an EGFP disruption assay (Supplementary Fig. 2b). The highest editing activity (near 80% EGFP-negative cells) was observed with a small Cas9 (1004 aa) derived from the poorly characterized bacterium *Collinsella* sp. AM34-10, which we named CoCas9. Interestingly, the *Collinsella* genus includes a rich variety of Cas9 orthologs (Fig. 1). CoCas9 shared the closest similarity with type II-C Cas9s from *Campylobacter jejuni*[13] and *Neisseria meningitidis*[6] (Fig. 2a) but with very low amino acid sequence identity (30.9% and 31.5%, respectively). The locus where CoCas9 was identified includes *cas1*, *cas2* and *cas9* genes and a CRISPR array composed of 24 spacer-direct repeat units (Fig. 2b). The tracrRNA sequence was identified between the CRISPR array and the *cas1* gene (see Methods), showing a predicted secondary structure resembling tracrRNAs of other type II systems[14], characterized by two stem loops located downstream from the crRNA anti-repeat region (Fig. 2c and Supplementary Data 1). CoCas9 recognizes a long, yet relaxed, PAM sequence (Fig. 2d). From the in vitro cleavage assay (see Methods) we obtained a frequency heatmap of the 256 four-nucleotide combinations contained in the PAM of CoCas9, revealing three main consensus sequences: 5'-$N_4$GWNT-3', 5'-$N_4$GCDT-3' and 5'-$N_4$ATDT-3' (W = A or T; D = A, G or T, Fig. 2e). Sanger sequencing of cleaved amplicons revealed that CoCas9 generates blunt-end

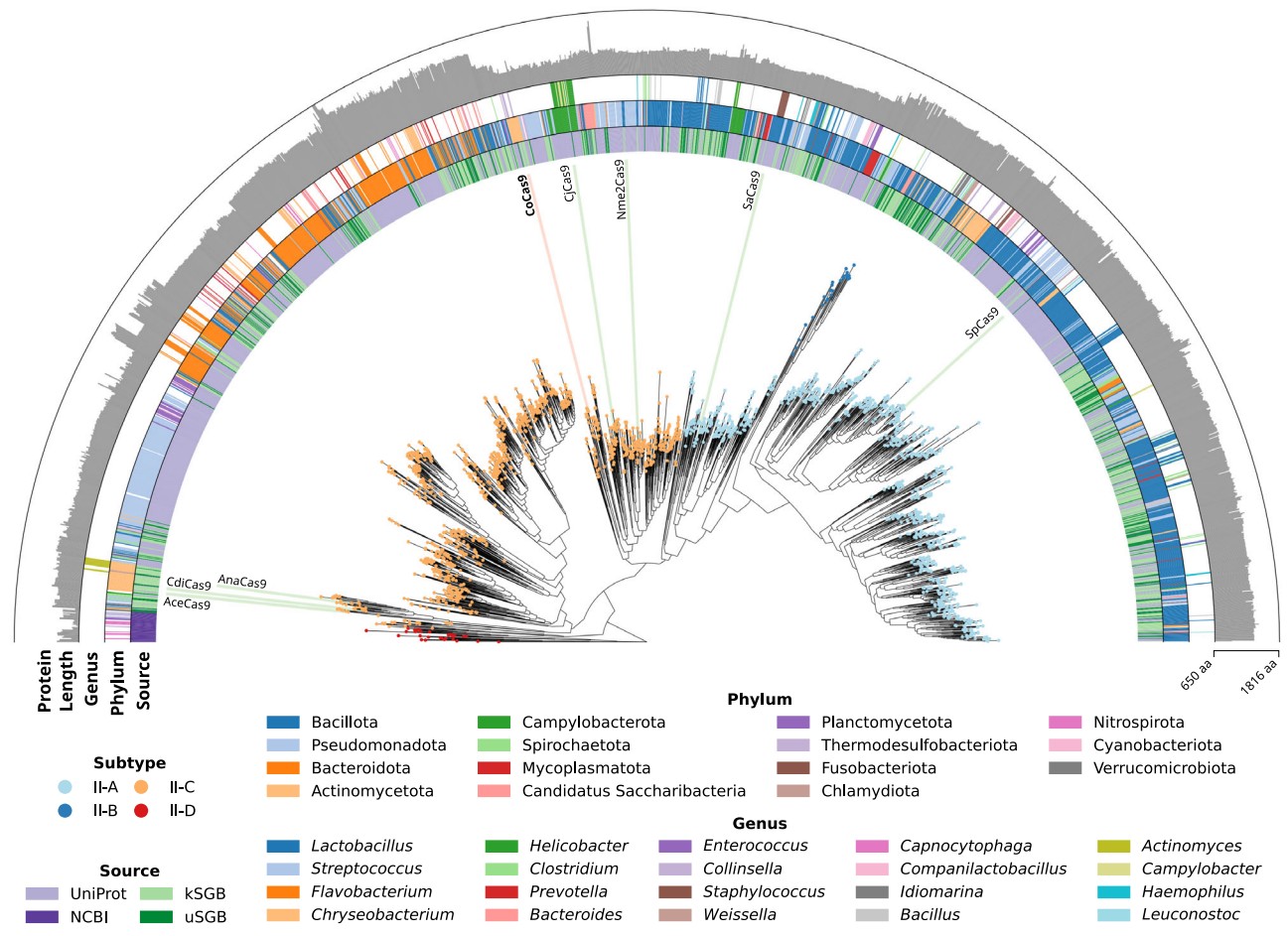

**Fig. 1 | Phylogenetic tree of Cas9 proteins.** Maximum likelihood phylogenetic tree of Cas9 proteins from known and unknown species-level genome bins (kSGB and uSGB, respectively) reconstructed from metagenomic data[8], together with Cas9 ortholog retrieved from UniProt and the NCBI nr database. Node color represents Cas9 subtypes. NCBI taxonomy is annotated for phyla with 10 or more members and for the 20 most abundant genera. Selected Cas9 orthologs mostly used for genome editing (SpCas9, SaCas9, Nme2Cas9 and CjCas9) or with experimentally determined three-dimensional structure (CdiCas9[58], AnaCas9[59] and AceCas9[60]) are annotated along the tree. The outer annotation ring represents Cas9 protein length. Source data are provided as a Source Data file.

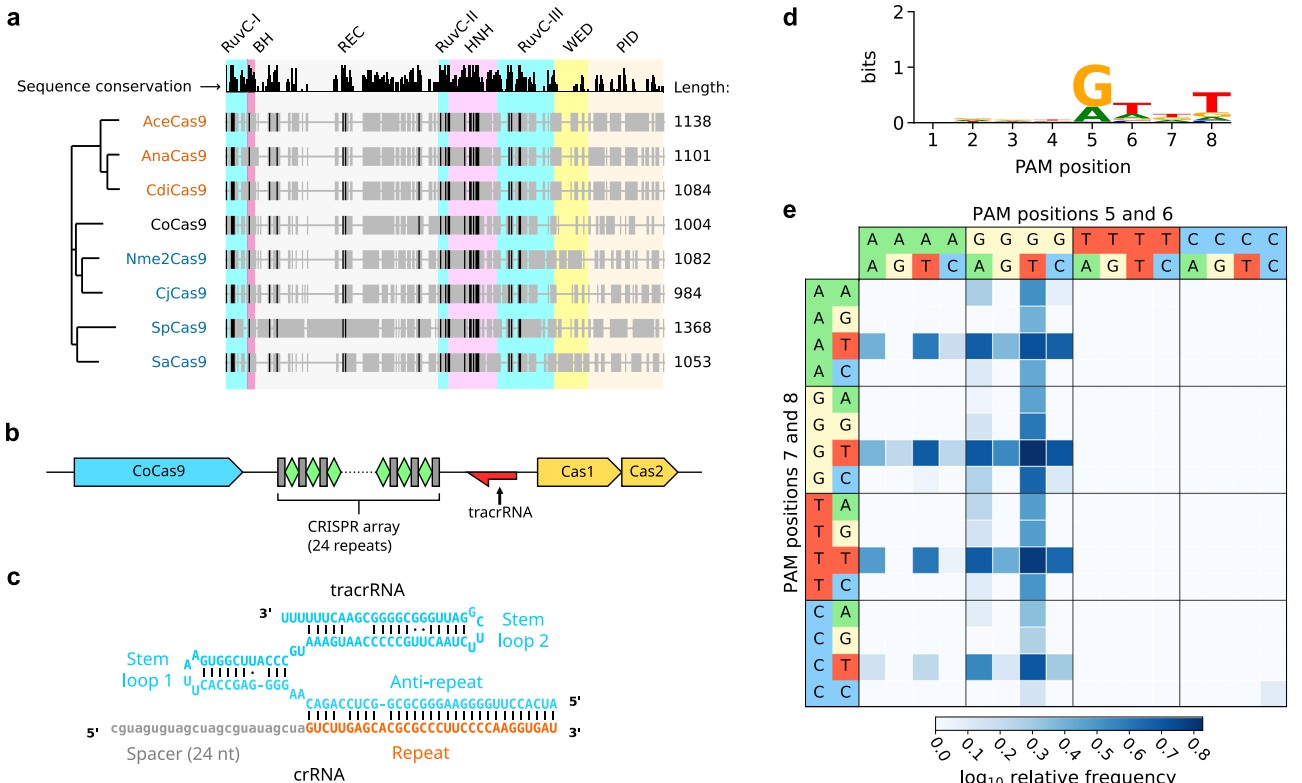

**Fig. 2 | CoCas9 is a compact nuclease from the Collinsella genus. a** Phylogenetic tree of selected Cas9 subfamilies mostly used for genome editing applications (blue) or with experimentally determined three-dimensional structure (orange). Protein alignments: gray, aligned protein sequences; black, conserved residues. Length: number of amino acids. **b** Scheme of the CRISPR locus where CoCas9 is located. **c** tracrRNA and crRNA predicted secondary structure. **d** Sequence logo representation of the PAM of CoCas9, determined using an in vitro cleavage assay. **e** Heatmap showing relative frequency of the 256 PAM nucleotide combinations at the four most informative positions, compared to the non-cleaved PAM library. Source data are provided as a Source Data file.

products, cleaving the target DNA 3 bp upstream of the PAM (Supplementary Fig. 3).

## CoCas9 is a highly active nuclease and base editor

The CoCas9 sgRNA was generated by fusing the crRNA + tracrRNA (Fig. 2c) through a GAAA tetraloop in a full scaffold (FS) sgRNA (Supplementary Data 1). Following previous works[5,15–17], the FS sgRNA was engineered to stabilize the secondary structure (FS-opt), to shorten the repeat:anti-repeat loop (TS), or both modifications were introduced simultaneously (TS-opt) (Supplementary Data 1). The engineered sgRNAs showed similar editing efficiency (Supplementary Fig. 4a) thus the TS-opt form was used hereafter. We then tested different spacer lengths, spanning from 22 to 24 nt, by targeting two genomic sites and noticed a small improvement with a 23 nt-long spacer at the *HBB* locus (Supplementary Fig. 4b). The efficiency of CoCas9 was then measured in HEK293T cells against a panel of 26 loci, showing variable editing levels, with up to 55% indels at specific targets (*HEKsite1* and *IL2RG*, Fig. 3a). To compare CoCas9 with the widely used SpCas9[4], we selected 24 loci having overlapping PAM and spacer sequences for the two orthologs (Supplementary Fig. 5a). CoCas9 produced comparable percentages of indels in the majority of the loci, albeit with reduced overall efficiency (mean difference 12.2%, Supplementary Fig. 5b). The expression levels of these two orthologs was verified to exclude editing variability depending on protein abundance (Supplementary Fig. 6).

To test the precision of CoCas9, we performed a genome-wide comparative off-target (OT) analysis with SpCas9 through GUIDE-seq[18]. To this aim, we selected four loci (*HPRT*, *VEGFAsite2*, *ZSCAN2* and *Chr6*) where both nucleases showed similar editing efficiency with overlapping spacer sequences (Supplementary Fig. 5a and Supplementary

Data 2). In all examined loci, CoCas9 produced considerably less OT cleavages than SpCas9 (Fig. 3b and Supplementary Fig. 7). The superior performance of CoCas9 was particularly striking at the OT benchmark locus *VEGFAsite2*[18,19], where CoCas9 produced 19-fold fewer OT sites than SpCas9 (101 and 1950 OT sites respectively, Supplementary Fig. 7b). Notably, at this locus, cleavages at the specific site by CoCas9 were 180-fold higher than those produced by SpCas9 (39.0% and 0.2% of GUIDE-seq reads respectively, Fig. 3b and Supplementary Fig. 7b).

We then tested CoCas9 base-editing by fusing its nickase version (carrying the D23A mutation in the RuvC-I domain) with the Tad-8e adenosine deaminase domain[20,21] to generate CoABE8e. In the 19 loci tested we detected up to 55% of A > G transition efficiency depending on the target locus (Fig. 3c). From a comparative analysis with the SpCas9-derived base editor[21] (SpABE8e), CoCas9 showed overall lower base-editing efficiency and a different window of activity at specific sites targeted by overlapping spacers (Supplementary Fig. 8).

## CoCas9 outperforms other small Cas9 orthologs

Thanks to its reduced length, CoCas9 is compatible with all-in-one AAV delivery. Currently, very few Cas9s with appreciable editing efficiency, mainly SaCas9[5] and Nme2Cas9[6], can be accommodated in these vectors together with their sgRNAs[6,22]. These orthologs were compared with CoCas9 by analyzing 11 genomic loci targeted by sgRNAs with overlapping spacers. We observed that in most cases (7 out of 11 loci) CoCas9 generated more indels than SaCas9, while Nme2Cas9 had overall lower activity compared to the other two nucleases (Fig. 4a). The specificity profiles of CoCas9 and SaCas9 were tested through GUIDE-seq by selecting five genomic loci where both orthologs showed similar activity (*TRAC*, *Chr6*, *TRBC*, *PDCD1*, *VEGFAsite3*) (Fig. 4a and Supplementary

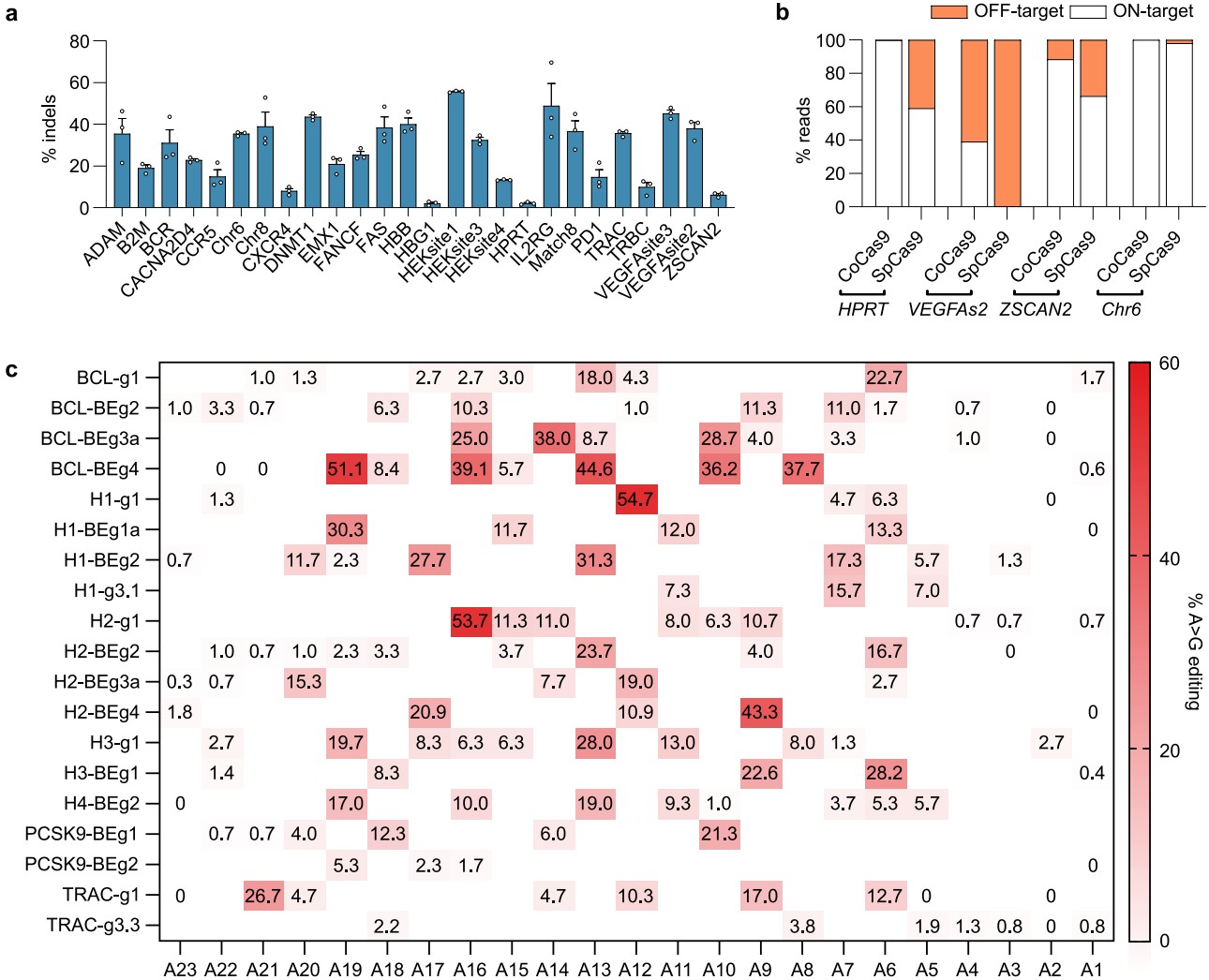

**Fig. 3 | CoCas9 nuclease and base editing activity. a** CoCas9 percentages of indels (TIDE analysis) in a panel of genomic loci in HEK293T cells. Data reported as mean ± SEM for $n = 3$ biologically independent experiments. **b** CoCas9 and SpCas9 on- versus off-target reads derived from GUIDE-seq in four genomic loci in HEK923T cells (Supplementary Fig. 7). **c** Summary of the frequency of A > G transitions by CoABE8e in HEK293T cells with sgRNAs targeting a panel of endogenous loci (Supplementary Data 2). Bases are numbered starting from the PAM proximal end of the guide. Data reported as the mean of $n = 3$ biologically independent experiments. Source data are provided as a Source Data file.

Fig. 9a). CoCas9 generated far less OTs than SaCas9 at two loci ($n = 3$ and $n = 15$ OT sites for *PDCD1* and $n = 0$ and $n = 31$ OT sites for *VEGFAsite3*, respectively) (Supplementary Fig. 9b), while nearly no OTs were observed for both nucleases in the remaining loci (Fig. 4b and Supplementary Fig. 9b–h). To deepen the comparison of CoCas9 with small Cas9 orthologs (SaCas9, Nme2Cas9 and CjCas9[13]), we included target sites with non-overlapping PAM and spacers ($n = 18$ sgRNAs for each ortholog) (Supplementary Fig. 10 and Supplementary Data 2). Overall, by considering both overlapping and non-overlapping guides, CoCas9 exhibited a mean editing efficiency of 30.3%, was significantly more active than Nme2Cas9 and CjCas9 (16.7% and 11.8% mean editing efficiency, respectively) and comparable to SaCas9 (28.1% mean editing efficiency) (Supplementary Fig. 10b). The targeting range of these orthologs was compared by analyzing the PAM availability in the human genome. The PAMs of CoCas9 span throughout the genome with a similar frequency as SpCas9 (11.8% and 10.0%, respectively), and can be found more frequently than PAMs of SaCas9 and Nme2Cas9 (2.9% and 9.9%, respectively), thus supporting the broad applicability of CoCas9 as an editing tool (Fig. 4d) (see Methods).

Small Cas9s are used to generate base editors with reduced molecular size[23]. We analyzed the editing profile of CoCas9 and SaCas9

derived base editors (CoABE8e and SaABE8e[21] respectively), having overlapping PAMs and spacers. The two base editors showed a similar editing profile, with CoABE8e more active at 3 out of 5 sites, while almost inactive at one locus (*TRAC*) (Fig. 4e and Supplementary Fig. 11). To extend the analysis to more base editors generated with small orthologs (Nme2Cas9, SauriCas9 and CjCas9)[23], we also analyzed sites targeted by non-overlapping PAMs. From the analysis of 9 sgRNAs targeting 3 genomic loci for each base editor we obtained very heterogeneous editing profiles (Supplementary Fig. 12a–c). The evaluation of the highest A > G conversion produced by each ortholog in the 3 examined loci showed overall similar activity among the analyzed base editors (Supplementary Fig. 12d). To estimate the therapeutic potential of CoABE8e, we retrieved 40,871 G > A pathogenic mutations from ClinVar[24] and predicted the fraction that could potentially be targeted by all tested base editors (see Methods). Notably, 10.2% of the considered mutations can be targeted solely by CoABE8e (Fig. 4f).

## Efficient editing of genomic loci using CoCas9 in clinically relevant models

To assess the potential of CoCas9 for therapeutic applications, we tested its activity in human primary cells. We transduced

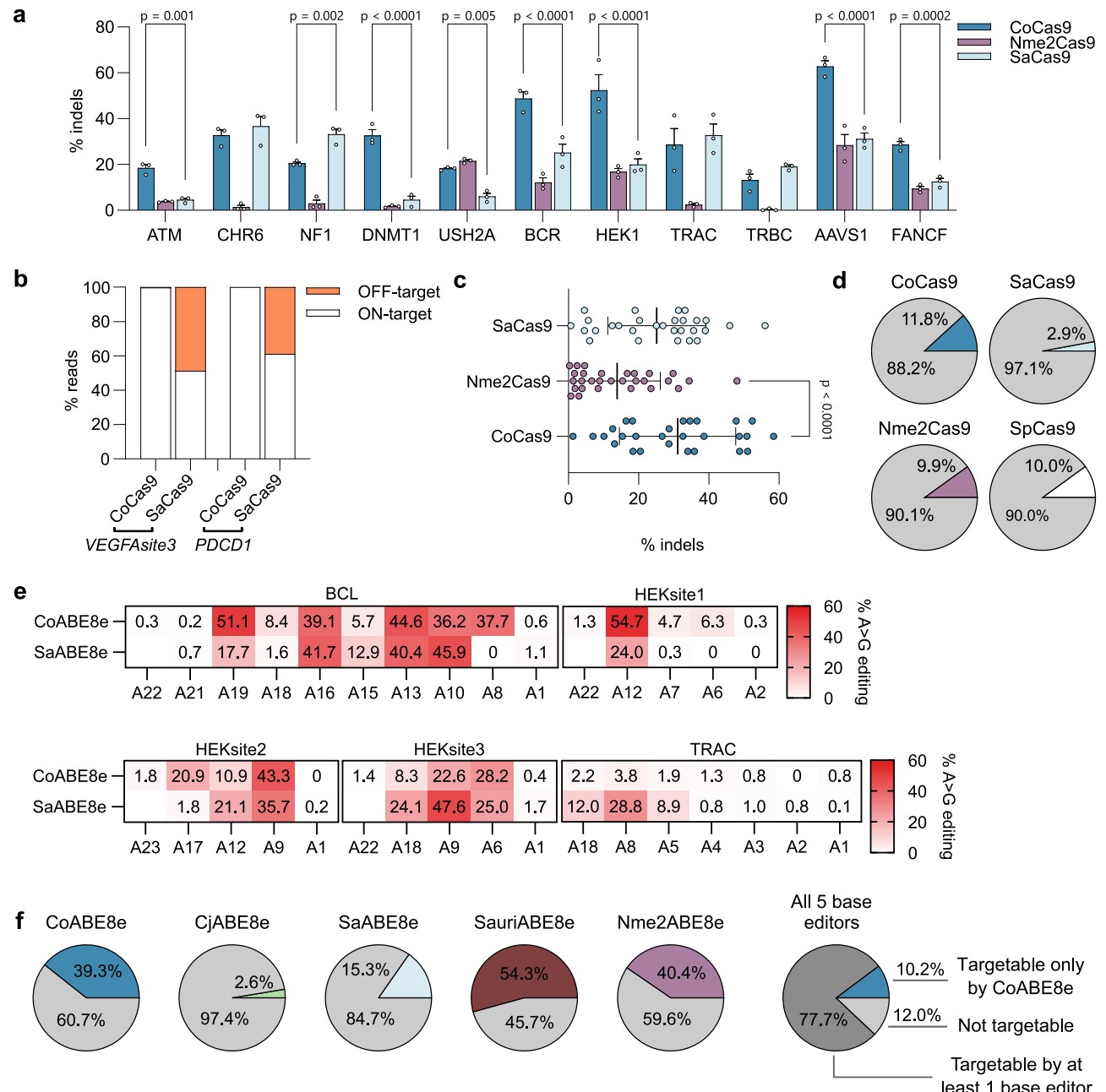

**Fig. 4 | CoCas9 is a highly active Cas9 ortholog with broad PAM compatibility.**
**a** Editing efficiency (% indels) of CoCas9, Nme2Cas9 and SaCas9. **b** Comparison of CoCas9 and SaCas9 percentages of on- versus off-target reads obtained by GUIDE-seq at *PDCD1* and *VEGFAsite3* loci. **c** Summary of data in (**a**) and Supplementary Fig. 10a. **d** Percentages of sites in the human genome targetable by the indicated compact Cas9s and SpCas9 (see Methods). **e** Heatmaps showing CoABE8e and SaABE8e base editing activity in 5 endogenous loci using overlapping sgRNAs and PAMs (Supplementary Fig. 11). **f** Editable fraction of human G > A pathogenic

mutations by the indicated compact ABE8es, either alone or in combination (last graph on the right) (see Methods). All experiments were performed in HEK293T cells. Statistical significance was assessed using a two-way ANOVA corrected using the Holm-Šídák method for each locus in panel (**a**) and a one-way ANOVA followed by a two-sided Holm-Šídák test in panel (**c**). In panel (**a**) data are shown as mean ± SEM, in panel (**c**) data are represented as mean ± SD; $n = 3$ biological replicates in panels (**a**, **e**); $n = 29$ biological replicates in panel (**c**). Source data are provided as a Source Data file.

hematopoietic stem/progenitor cells (HSPCs), human bronchial epithelial cells (HBE) and human skin fibroblasts (HSF) with lentiviral vectors and evaluated the editing activity in 6 loci after puromycin selection (Fig. 5a–c). CoCas9 showed high editing activity in all tested genomic loci of all primary cell types tested, with the exception of a single locus (TRAC) in HSPCs showing reduced indels (~10%, Fig. 5a–c).

The promising properties of CoCas9 led us to test the delivery of the derived base editor through an all-in-one AAV (AAV-v1 and -v2 schematized in Supplementary Fig. 13a). AAV transduction of CoABE8e

with the sgRNA targeting *HEKsite2* showed high percentages of A > G transitions, with AAV-v1 reaching 68% of base editing (Fig. 5d), thus improving the activity obtained via plasmid transfection (53.7%, Fig. 3c).

To test the editing efficiency of CoCas9 nuclease delivered through all-in-one AAV, we targeted the *RHO* gene, which is mutated in a common form of autosomal dominant retinitis pigmentosa[25–27]. Two different configurations, AAV-v3 and AAV-v4 (Supplementary Fig. 13b), reached up to 30% of editing efficiency in HEK293 stably expressing a human *RHO* (h*RHO*)-EGFP minigene (Fig. 5e). To assess the in vivo

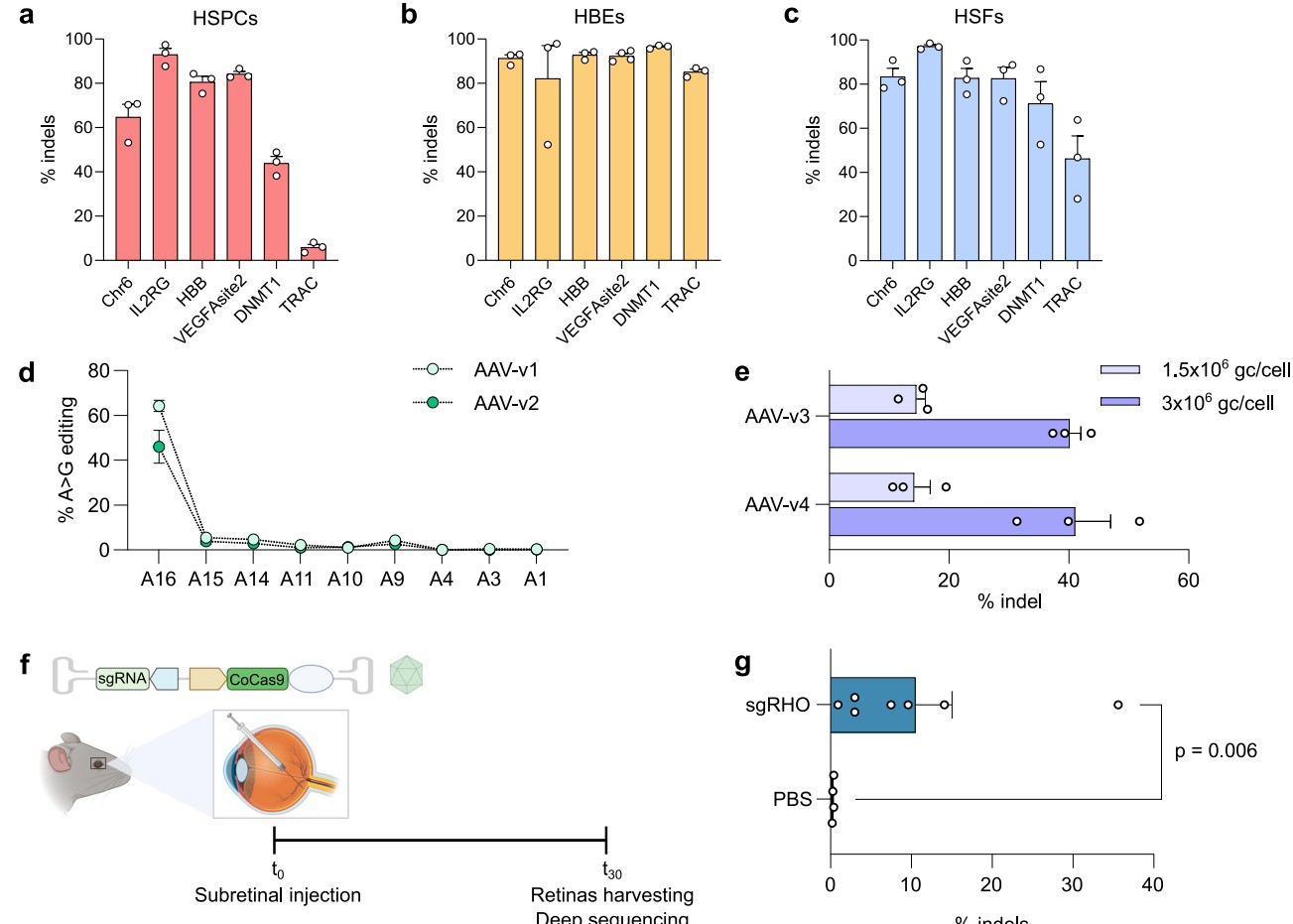

**Fig. 5 | CoCas9 is active in human primary cells and the murine retina.** Editing efficiency (% indels) of CoCas9 in 6 genomic loci of hematopoietic stem/progenitor cells (HSPCs) (representative experiment) (**a**), bronchial epithelial cells (HBE) (**b**) and skin fibroblasts (HSFs) (**c**) after transduction with lentiviral vectors and puromycin selection. **d** Frequency of A > G transitions at the HEKsite2 locus in HEK293T cells transduced with AAV-v1 and AAV-v2 (7 × 10⁵ genome copies/cell, gc/cell). **e** Percentages of indels at the *RHO* locus generated by AAV-v3 and AAV-v4 transduction in HEK293-*RHO*-EGFP cells with two amounts of gc/cell. **f** Flowchart of AAV8-CoCas9 delivery in murine retinas. Created with BioRender.com. (**g**) Editing efficiency of AAV8-CoCas9 injected in the mouse retina. Statistical significance was assessed using a two-sided t-test in panel (**g**). In panel (**a**–**e**, **g**) data are shown as mean ± SEM; $n = 3$ biological replicates in panels (**a**–**e**), $n = 7$ (sgRHO) and $n = 4$ (PBS) independent retinas in panel (**g**). Source data are provided as a Source Data file.

editing activity of CoCas9, we exploited the design of AAV-v3 (Fig. 5f and Supplementary Fig. 13b) to deliver CoCas9 in a knock-in mouse model harboring a single copy of the h*RHO* gene (see Methods). We administered an all-in-one AAV8 with CoCas9 targeting h*RHO* to 5 weeks-old mice via subretinal injection (Fig. 5f). Four weeks after the administration we observed up to 35.6% of editing efficiency at the h*RHO* target, with a mean of 10.5% in 7 retinas (Fig. 5g). These results demonstrated the potential of CoCas9 for clinical development.

## Discussion

Through the interrogation of a massively expanded database of microbial genomes from the human microbiome, we uncovered a large reservoir of Cas9 orthologs with compelling properties for genome editing. By selectively searching for low molecular weight Cas9s, we obtained orthologs that are compatible with AAV vectors, thus overcoming the necessity of using dual AAVs to deliver the Cas9 and its sgRNA. Among the identified CRISPR loci we found CoCas9, a compact (1004 aa) and efficient high-fidelity Cas9 with a complex PAM sequence (5'-N₄GWNT-3', 5'-N₄GCDT-3' and 5'-N₄ATDT-3') that, despite its length, matches 11.8% of sites in the human genome, a frequency comparable to that of SpCas9 (10.0% of sites). By comparing CoCas9 with other small Cas9 orthologs, namely SaCas9, Nme2Cas9, and

CjCas9, we showed that CoCas9 exhibits an editing activity similar to SaCas9 and significantly higher than Nme2Cas9 and CjCas9. We then compared the specificity of CoCas9 and SaCas9, demonstrating that CoCas9 is more precise than SaCas9, despite the latter's narrow PAM range (2.9% of sites in the human genome), which is expected to reduce the number of off-target events. Thus, despite their similarity in on-target activity, CoCas9 is preferable to SaCas9 for its intrinsic specificity and targeting range. Direct comparison of the specificity of CoCas9 with Nme2Cas9, which is also highly precise[6], was not feasible due to the incompatibility of their PAM sequences.

The advantage of type II Cas nucleases compared to type V is the opportunity to generate efficient base editors through the inactivation of one out of the two catalytic domains. We found that CoABE8e has an efficient base editing activity, similar to other base editors derived from small Cas9 orthologs, thus extending the targeting range of current tools. However, the wide editing window of CoABE8e could potentially correspond to a high rate of unwanted bystander edits, which is a limitation shared by all ABE8e base editors[23,28]. The recently developed ABE9 base editors[28], which exhibit a much narrower editing window, could be used to overcome this issue.

Overall, by demonstrating the high activity of CoCas9 in clinically relevant models such as HSPCs and the murine retina, we have shown

that the natural diversity of Cas9 orthologs identified from metagenomic data can expand the genome editing toolbox and has the potential to address the complexity of gene therapy applications. Moreover, a potential advantage of Cas9 orthologs identified from commensal species in the human microbiome, in contrast to orthologs from pathogenic species (e.g., *Streptococcus pyogenes*), could be a reduced pre-existing adaptive immunity, which should be investigated in future studies. Finally, structure-based engineering and directed evolution approaches have successfully generated a plethora of engineered Cas9 variants with enhanced activity, specificity, or a relaxed targeting range[29]. While this study focuses on comparing CoCas9 with naturally occurring orthologs, its performance in comparison to engineered variants remains to be assessed. In this regard, CoCas9 represents a valuable candidate for optimization through engineering approaches, warranting further investigation.

## Methods

### Ethical statement

All mouse experiments were performed in accordance with the Association for Research in Vision and Ophthalmology Statement for the Use of Animals in Ophthalmic and Vision and with the Italian Ministry of Health regulation for animal procedures (Ministry of Health authorization number: 252/2022-PR).

For experiments involving human adult HSPCs, written informed consent was obtained from all subjects. All experiments were performed in accordance with the Declaration of Helsinki. The study was approved by the regional investigational review board (reference DC2022-5364, CPP Île-de-France II "Hôpital Necker Enfants malades", Paris, France).

### Identification of Cas9 orthologs and tracrRNA prediction from metagenomic data

*cas1*, *cas2* and *cas9* genes were identified from the protein annotation of 154,723 bacterial and archaeal metagenome-assembled genomes (MAGs), reconstructed from the human microbiome[8]. CRISPR arrays were identified using MinCED[30] version 0.4.2 (with default parameters). Only loci having a CRISPR array and *cas1-2-9* genes at a maximum distance of 10 kbp from each other were considered. Loci containing Cas9 proteins shorter than 950 aa were discarded. The resulting 17,173 CRISPR-Cas9 loci were filtered by selecting short proteins (less than 1100 aa) from poorly characterized species. Cas9 proteins from the same species, having similar length but slightly different sequence, were compared by multiple sequence alignment. Proteins presenting deletions in nuclease domains were discarded. The remaining proteins were compared for sequencing coverage and the ortholog with the highest coverage was selected for each species. Cas9 subtype was predicted using CRISPRCasTyper v1.2.1[31].

Prediction of tracrRNAs for CRISPR-Cas9 loci of interest was performed based on the work by Chyou and Brown[32]. Starting from unique direct repeats in the CRISPR array, BLAST version 2.2.31 (with parameters -task blastn-short -gapopen 2 -gapextend 1 -penalty −1 -reward 1 -evalue 1 -word_size 8) was used to identify anti-repeats within a 3000 bp window flanking the CRISPR-Cas9 locus. RNIE[33] version 0.01 was used to predict Rho-independent transcription terminators (RITs) near anti-repeats. Putative tracrRNA sequences, starting with an anti-repeat and ending with either a RIT (when found) or a poly-T, were combined with direct repeats to form sgRNA scaffolds. The secondary structure of sgRNA scaffolds was predicted using RNAsubopt version 2.4.14 (with parameters --noLP -e 5)[34]. sgRNAs lacking the functional modules identified by ref. [14] namely the repeat:anti-repeat duplex, nexus and 3' hairpin-like folds, were discarded.

The identified Cas9 orthologs coding sequences were modified by the addition of an SV5 tag at the N-terminus and two nuclear localization signals (1 at the N-term and 1 at the C-term) and were human codon-optimized (Supplementary Data 3). The constructs for the coding sequences, as well as the sgRNA scaffolds, were obtained as synthetic fragments from either Genscript or Genewiz.

### Cas9 phylogenetic tree

17,137 Cas9 sequences were first clustered at 70% identity and 50% coverage using MMseqs2[35] version 13.45111, together with 2076 Cas9 sequences retrieved from UniProt release 2020_06, obtaining 567 clusters. Proteins with lengths more than two standard deviations below the median length of each cluster were excluded to eliminate partial or fragmented sequences. 49 additional subtype II-D Cas9 orthologs were retrieved from the NCBI nr database using 4 iterations of PSI-BLAST[36], starting previously reported II-D orthologs[37]. The resulting 18,393 sequences were clustered at 90% identity and 90% coverage using MMseqs2 version 13.45111, obtaining 2206 clusters. The centroid sequences were aligned using mafft[38] version 7.490 (with parameters –maxiterate 10). The resulting alignment was used to generate a phylogenetic tree using IQ-TREE[39] version 2.0.3 with automatic model selection and 1000 bootstrap replicates. The resulting tree was rooted on the split between subtypes II-C and II-D, and displayed using GraPhlAn[40] version 1.1.3.

### Construction of the randomized PAM library

The randomized PAM library was prepared as described in ref. [41]. Briefly: one synthetic DNA oligonucleotide containing an EcoRI site and a 8-nt randomized sequence (top oligo) was obtained from Eurofins, together with another DNA oligo that anneals to the 3' region flanking the randomized sequence leaving an SphI-compatible end (bottom oligo). The bottom strand of the annealed oligo duplex was filled-in by Klenow(exo-) incubation and digested with EcoRI for ligation into a SphI/EcoRI-cut p11-lacY-wtx1 vector (Addgene Plasmid #69056). The ligation product was then electroporated into MegaX DH10B T1R Electrocomp™ Cells (Thermo Fisher Scientific) to reach a theoretical library coverage of 100X. Colonies were harvested and the plasmid DNA was purified by maxi-prep (Macherey-Nagel). Two PCR steps (Phusion HF DNA polymerase - Thermo Fisher Scientific) were performed to prepare the plasmid PAM library for NGS analysis: the first, using a set of forward primers and two different reverse primers, to amplify the region containing the protospacer and the randomized PAM and the second to attach the Illumina Nextera DNA indexes and adapters (Supplementary Data 4). PCR products were purified using Agencourt AMPure beads in a 1:0.8 ratio. The library was analyzed with a 150 bp single read sequencing, using a v3 flow cell on an Illumina MiSeq sequencer.

### In vitro assay for PAM identification

The in vitro PAM evaluation of the selected Cas9 orthologs was performed according to the protocol from Karvelis et al.[42] with few modifications. In brief: the synthetic DNA encoding the human codon optimized version of the Cas9 genes was obtained from Genscript and cloned into an expression vector for in vitro transcription and translation (IVT) (pT7-N-His-GST Thermo Fisher Scientific). Reaction was performed according to the manufacturer protocol (1-Step Human High-Yield Mini IVT Kit - Thermo Fisher Scientific). The Cas9-guideRNA ribonucleoprotein (RNP) complex was assembled by combining 20 μL of the supernatant containing soluble Cas9 protein with 1 μL of Ribo-Lock RNase Inhibitor (Thermo Fisher Scientific) and 2 μg of guide RNA. The RNP was used to digest 1 μg of the randomized PAM plasmid DNA library for 1 h at 37 °C.

A double stranded DNA adapter[42] (Supplementary Data 4) was ligated to the DNA ends generated by the targeted Cas9 cleavage and the final ligation product was purified using a GeneJet PCR Purification Kit (Thermo Fisher Scientific).

One round of a two-step PCR (Phusion HF DNA polymerase - Thermo Fisher Scientific) was performed to enrich the sequences that

were cut using a set of forward primers annealing on the adapter and a reverse primer designed on the plasmid backbone downstream of the PAM (Supplementary Data 4). A second round of PCR was performed to attach the Illumina indexes and adapters. PCR products were purified using Agencourt AMPure beads in a 1:0.8 ratio. The library was analyzed with a 71 bp single read sequencing, using a flow cell v2 micro, on an Illumina MiSeq sequencer.

PAM sequences were extracted from Illumina MiSeq reads and used to generate PAM sequence logos, using Logomaker version 0.8. PAM heatmaps[43,44] were used to display PAM enrichment, computed dividing the frequency of PAM sequences in the cleaved library by the frequency of the same sequences in a control uncleaved library.

### Estimation of PAM frequency and editable adenines
The number of targetable genomic sites was computed counting the number of occurrences of PAMs in the hg38 human reference genome for each Cas9 nuclease. For base editors, mutations in the ClinVar database[24] were filtered to select G > A and C > T single nucleotide variants (SNVs) annotated as pathogenic or likely pathogenic, for a total of 40,871 SNVs. Editing windows of base editors were estimated as positions showing at least 10% mean editing on all tested guides (Supplementary Data 5). Selected SNVs were considered targetable if the surrounding sequence contained a PAM, on the appropriate strand, that would place the mutated adenine inside the estimated editing window.

### CoCas9 cleavage pattern assay
In vitro CoCas9 cleavage products were analyzed by PCR and Sanger sequencing. EGFP and B2M PCR products were obtained amplifying pEGFP-N1 and HEK293T DNA respectively (HotFire polymerase, Solis BioDyne), using primers indicated in Supplementary Data 6. CoCas9 protein was produced using the 1-Step Human High-Yield Mini IVT Kit (Thermo Fisher Scientific), while *EGFP* and *B2M* sgRNAs were in vitro transcribed (HighYield T7 RNA Synthesis Kit, Jena Bioscience) starting from the amplification (Phusion HF DNA polymerase - Thermo Fisher Scientific) of the plasmid bearing CoCas9 sgRNA (primers reported in Supplementary Data 6). The RNP complex was assembled using 300 nM sgRNA, 3.8 µl of in vitro translated CoCas9 and RiboLock RNase Inhibitor 1 U/µl (Thermo Fisher Scientific) in 10X nuclease reaction buffer (200 mM HEPES, 1 M NaCl, 50 mM MgCl₂, 1 mM EDTA). After 10 min of incubation, 30 nM of corresponding PCR was added and the reaction (30 µl) was left for 1 h at 37 °C; 1 µl of RNase A/T1 Mix (Thermo Fisher Scientific) and 0.7 µg/µl of Proteinase K (Thermo Fisher Scientific) were finally added. The PCR fragments were loaded on agarose gel, purified and sent for Sanger sequencing (Eurofins) using primers used for amplification.

### Mammalian expression plasmids and constructs
A pX330-derived plasmid was used to express the Cas9 orthologs and relative sgRNA in mammalian cells. Briefly, pX330 was modified by substituting the SpCas9 and its sgRNA scaffold with the human codon-optimized sequence of the orthologs and its sgRNA scaffold (either full length or trimmed). The Cas9s coding sequences and the sgRNA scaffolds were obtained as synthetic fragments from either Genscript or Genewiz; Cas9s were fused with a V5 tag at the N-terminus and two nuclear localization signals (1 at the N-term and 1 at the C-term). Spacer sequences were cloned into the pX-Cas9 plasmids as annealed DNA oligonucleotides containing a variable 20 or 24-nt spacer sequence using double BsaI sites present in the plasmid. The list of spacer sequences used in the EGFP disruption assay and in the endogenous loci targeting is reported in Supplementary Data 2.

For base editor constructs, the nCoCas9 (D23A) obtained by PCR mutagenesis was fused with the TadA-8e domain from CP1041-ABE8e (Addgene #138493) and introduced in a pCMV-derived plasmid, while

the optimized trimmed version of CoCas9 sgRNA (CoCas9 TS-opt) was cloned in a pUC19-derived plasmid (Supplementary Data 1).

pAAV-v1 and pAAV-v2 (AAV-v1 and AAV-v2 plasmids) were synthesized by Vectorbuilder; gRNA1_CoCas9_HEKsite2 (shortened H2-g1) was selected for pAAV-v1 and pAAV-v2 after transfection experiments shown in Fig. 3c and Supplementary Fig. 12c. pAAV-v3 and pAAV-v4 (AAV-v3 and AAV-v4 plasmids) were obtained by adding the U6-gRNA3.1_CoCas9_RHO (Supplementary Data 2) cassette and substituting EGFP with the CoCas9 sequence in pTIGEM[45]. Final constructs are shown in Supplementary Fig. 13.

A plentiCRISPR v1 plasmid (Addgene Plasmid 49535) was used to express Cocas9 in human primary cell lines. Briefly, plentiCRISPR v1 was modified by substituting the SpCas9 and its sgRNA scaffold with the human codon-optimized sequence of CoCas9 and its sgRNA scaffold. Spacer sequences were cloned in the plentiCRISPR v1 using BsmBI restriction sites.

### Cell lines, primary human cells and culture conditions
HEK293T cells (obtained from ATCC, #CRL-11268), U2OS.EGFP cells (a kind gift of Claudio Mussolino, University of Freiburg), harboring a single integrated copy of an EGFP reporter gene, AAVpro-293T (obtained from Takara, #632273), HEK293-RHO-EGFP cells and HSF (from the Coriell Institute, GM05659) were cultured in DMEM (Life Technologies) supplemented with 10% FBS (Life Technologies), 2 mM GlutaMax (Life Technologies) and penicillin/streptomycin (Life Technologies). HBE (BE121 from the Italian Cystic Fibrosis Research Foundation) were cultured in LHC9/RPMI 1640 medium (1:1) without serum[46,47] (provided by the Italian Cystic Fibrosis Research Foundation[48]), supplemented with rho-associated protein kinase 1 inhibitor (Y-27632, 5 µM) (Merck) and SMAD-signaling inhibitors, bone morphogenetic protein antagonist (DMH-1, 1 µM), and transforming growth factor β antagonist (A 83-01, 1 µM) to promote basal cell proliferation[49]. HEK293-RHO-EGFP cells were obtained by stable transfection of HEK293 cells (obtained from ATCC, #CRL-1573) with a pcDNA5/TO-RHO-EGFP reporter plasmid, obtained by cloning a fragment of the *RHO* gene up to exon 2 (retaining introns 1 and 2) fused to part of RHO cDNA containing exons 3–5 in frame with the EGFP coding sequence into a pCDNA5/TO expression plasmid. Cells were pool-selected with 5 µg/ml Hygromycin (Invivogen) and single clones were subsequently isolated and expanded. All cells were incubated at 37 °C and 5% CO₂ in a humidified atmosphere. All cells tested mycoplasma negative (PlasmoTest, Invivogen).

Non-mobilized human adult HSPCs were obtained from patients with sickle cell disease and mobilized human adult HSPCs from healthy donors. Written informed consent was obtained from all subjects. All experiments were performed in accordance with the Declaration of Helsinki. The study was approved by the regional investigational review board (reference DC2022-5364, CPP Île-de-France II "Hôpital Necker Enfants malades", Paris, France). HSPCs were purified by immunomagnetic selection (Miltenyi Biotec) after immunostaining using the CD34 MicroBead Kit (Miltenyi Biotec). HSPCs cells were thawed and cultured for 24 h at a concentration of 106 cells/ml in pre-activation medium (PAM) composed of X-VIVO20 supplemented with penicillin/streptomycin (Gibco) and recombinant human cytokines: 300 ng/ml hSCF, 300 ng/ml Flt-3L, 100 ng/ml TPO, 20 ng/ml interleukin-3 (IL-3) (PeproTech), and 10 mM SR1 (STEMCELL Technologies). After pre-activation, cells (106 cells/ml) were cultured in PAM supplemented with 10 mM PGE2 (Cayman-Chemical) on RetroNectin coated plates (10 mg/cm², Takara Bio) for at least 2 h.

### Cell line transfections
For EGFP disruption assay 200,000 U2OS.EGFP cells were nucleofected with 1 µg of px-Cas9 plasmids bearing a sgRNA designed to target EGFP using the 4D-Nucleofector™ X Kit (Lonza), DN100

program, according to the manufacturer's protocol. After electroporation, cells were seeded in 96-well plates and transferred to 24-well plates after 48 h for growth expansion.

For editing analyses of genomic loci, 100,000 HEK293T cells were seeded in a 24-well plate 24 h before transfection. Cells were then transfected either with 1 μg of pX-Cas-sgRNA plasmid expressing both the Cas9s and the sgRNAs, or with 500 ng of a pX-Cas9 plasmids and 250 ng of a pUC-sgRNA constructs, using the TransIT-LT1 reagent (Mirus Bio) according to the manufacturer's protocol. Cell pellets were collected 3 days post-transfection for indel evaluations. For base editing experiments, cells were co-transfected as described above with 750 ng of pCMV-ABE8e containing the specific Cas9s and 250 ng of pUC-sgRNA.

## AAV production
For AAV-DJ, AAVpro-293T cells ($10^7$ cells/dish in three P150 dishes) were PEI transfected with pHelper, pAAV Rep-Cap and the proper pAAV-v1-4 construct (Supplementary Data 7). Three days post-transfection, media and cells were collected, centrifuged and processed separately. Cells were washed and lysed with an acidic citrate buffer (55 mM citric acid, 55 mM sodium citrate, 800 mM NaCl, pH 4.2)[50]. The lysates were cleared by centrifugation, treated with HEPES buffer 1 M, DNaseI and RNaseA (Thermo Fisher Scientific) and then mixed with the collected medium and 500 mM NaCl. AAVs were precipitated with 8% polyethylene glycol (PEG) 8000 overnight at 4 °C, then collected by centrifugation and resuspended in TNE Buffer (100 mM Tris·Cl, pH 8.0, 150 mM NaCl, 20 mM EDTA) followed by 1:1 chloroform extraction. AAV titration was performed by qPCR following the Addgene protocol[51] (https://www.addgene.org/protocols/aav-titration-qpcr-using-sybr-green-technology/).

AAV8 vectors were produced by InnovaVector S.R.L. as previously described in ref. 52. Briefly, for each large vector preparation, a suspension of $2.2 \times 10^9$ low-passage HEK293 cells was triple-transfected by calcium phosphate with 1,000 μg of helper, 520 μg of packaging, and 520 μg of pAAV cis-plasmids and plated in CellSTACK-10 (Corning, Amsterdam, The Netherlands). The following day, the medium was changed with serum-free DMEM and cells were harvested 72 hr after transfection. Cells were lysed by three rounds of freeze–thaw to release AAV8 particles. The lysate was then incubated with both DNase I (8000 U for large vector preparation) and RNase A (200 U for large vector preparation) (Roche Diagnostics, Monza, Italy) for 30 min at 37 °C to get rid of nucleic acids and with 10% Octyl-βD-glucopyranoside (Sigma-Aldrich, St. Louis, MO) to complete lysis. AAV vectors were then purified by two sequential $CsCl_2$ gradients. Gradient fractions were measured by refractometry, and those with refractive index ranging between 1.3660 and 1.3740 were pooled. All fractions were desalted in phosphate-buffered saline. Glycerol was added to the concentrated AAV lots to a final concentration of 5% (v/v), and the preparations were aliquoted and stored at −80 °C. The titer (genome copies/ml) of each viral preparation was determined by averaging the titer achieved by dot-blot analysis and by qPCR quantification using TaqMan (Applied Biosystems, California, USA)[52].

## AAV transduction in cell lines
For AAV-DJ transduction in cell lines $10^5$ cells were transduced in a 24-well plate. After 3 days the medium was changed to remove the vector and cells were collected 6 days post-transduction for editing analysis.

## Lentiviral vectors production and cell transduction
Lentiviral vectors were produced in HEK293T cells at 80% confluency in p150 plates. 25 μg of transfer vector (lentiCRISPR v1) plasmid, 7.5 μg of VSV-G and 18.8 μg of Δ8.91 packaging plasmid were transfected using PEI. The culture medium was replaced the day after with complete DMEM. The viral supernatant was collected after 48 h and filtered through a 0.45 μm PES filter. Lentiviral vectors were concentrated and

purified with 20% sucrose cushion by ultracentrifugation for 2 h at 4 °C and 150,000 × g. Pellets were resuspended in OptiMEM and aliquots stored at −80 °C. Vector titres were measured as reverse transcriptase units (RTU) by SG-PERT method[53].

For transduction experiments, HSF and HBE were seeded (60,000 cells/well and 100,000 cells/well, respectively) in 12-well plates and the day after were transduced with 2 RTU of lentiviral vectors. 72 h later, cells were cultured with puromycin (2 μg/ml) and collected 10 days from transduction for editing analysis. HSPCs ($10^6$ cells/ml) were transduced in RetroNectin coated plates in PAM supplemented with 10 mM PGE2, protamine sulfate (4 mg/ml, ProtamineChoay), and Lentiboost (1 mg/ml, Sirion Biotech) using 5.5–6.5 RTU of lentiviral vector. 24 h after transduction cells were cultured in basal erythroid medium (BEM) supplemented with hSCF, IL-3, EPO (Eprex, Janssen-Cilag), and hydrocortisone (Sigma) and after additional 24 h were cultured with puromycin (1.8 ug/ml). Collection occurred on day 6 from transduction.

## Animal model
Mice were housed at TIGEM animal house (Pozzuoli, Italy). The knock-in mouse model used in this study was kindly provided by Theodore G. Wensel[54]. This mouse model harbors the full-length human rhodopsin (h*RHO*) gene including the P23H mutation (hRHO-P23H). hRHO-P23H C-terminus is fused to the red fluorescent protein (RFP) coding sequence (hRHO-P23H-RFP). hRHO-P23H-RFP −/− mice were maintained by crossing homozygous females and males. Experimental heterozygous hRHO-P23H-RFP +/− animals were obtained by crossing homozygous hRHO-P23H-RFP −/− with C57BL/6 J mice obtained from Envigo Italy SRL (Udine, Italy). Genotype analysis was performed by sequencing genomic DNA from the knock-in mouse as previously described in ref. 54. Briefly, the following genotyping PCR primers were used for the detection of the hRHO-P23H-RFP knock-in allele using 50 ng of genomic DNA: the forward primer was designed on the exon 5 sequence (5′-GTTCCGGAACTGCATGCTCACCAC-3′) while the reverse primer was designed on the 3′-UTR hRHO sequence (5′-GGCGCTGCTCCTGGTGGG-3′). The genotyping PCR generated a 975 bp knock-in band and a 194 bp WT band in heterozygous mice, a single 975 bp knock-in band in homozygous mice and a single 194 bp band in WT mice. The expression of the hRHO-P23H-RFP fusion was verified by fluorescence microscopy of retinas and by immunoblotting.

## Subretinal AAV injections in mice
Subretinal injections with AAV8 (AAV-v3) were performed on the temporal side of the eye using 1 μl of vector-containing solution via a trans-scleral, trans-choroidal approach[55] in accordance with both the Association for Research in Vision and Ophthalmology Statement for the Use of Animals in Ophthalmic and Vision and with the Italian Ministry of Health regulation for animal procedures (Ministry of Health authorization number: 252/2022-PR). The surgical intervention was performed under general anesthesia; mice received an intraperitoneal injection of ketamine (10 mg/Kg) combined with medetomidine (1 mg/Kg). hRHO-P23H-RFP +/− mice were injected sub-retinally at around 5-weeks of age with a combination of two AAV8 vectors: one expressing CoCas9 (AAV-v3, $2 \times 10^9$ genome copies/eye) and the other the enhanced green fluorescent protein (EGFP, $2 \times 10^8$ genome copies/eye, to track the injected area) both under the control of the ubiquitous CMV promoter. Control eyes received the AAV8 CMV-EGFP vector alone. Right and left eyes were randomly assigned to each treatment group. One month post-injection mice were euthanized, the neural retina of each eye was collected and separated into EGFP+ and EGFP-portions for indel formation analyses.

## Evaluation of editing efficiency
For EGFP disruption assay, EGFP fluorescence was measured four days after nucleofection using a BD FACSCanto (BD) flow cytometer.

DNA was extracted using the QuickExtract DNA Extraction Solution (Lucigen) for cell culture 3 days after transfection, 6 days after AAV transduction and 10 days after lentiviral transduction. DNeasy Blood & Tissue kit was used to isolate genomic DNA from cells 10 days after transduction and from mouse retina samples at 30 days from AAV transduction according to the manufacturer's instructions. As for HSPCs, genomic DNA was extracted from control and edited cells using PURE LINK Genomic DNA Mini kit (Life Technologies).

To amplify the target loci, PCR reactions were performed using the HOT FIREPol DNA polymerase (Solis BioDyne), using the oligonucleotides listed in Supplementary Data 6. The amplified products were purified, Sanger sequenced (Microsynth) and analyzed with the TIDE web tool version 3.30 (http://shinyapps.datacurators.nl/tide/) to quantify indels or with the EditR web tool version 1.0.0 (http://baseeditr.com) to quantify base editing events.

In vivo editing efficiency of CoCas9 at the hRHO gene target in the mice retinas was analyzed by deep sequencing. To amplify the target locus, a first round of PCR reactions was performed using HOT FIREPol DNA polymerase (Solis BioDyne), using 100 ng of DNA as template and the oligonucleotides listed in Supplementary Data 4. The amplified products were purified (Agencourt AMPure beads), quantified and used for a second round of PCR (Phusion HF DNA polymerase - Thermo Fisher Scientific) to attach the Illumina indexes. After final purification and quantification, a pool of each sample with the same nM was analyzed with a 150 bp single read sequencing, using a v2 flow cell on an Illumina MiSeq sequencer.

### Off-target analysis

GUIDE-seq experiments were performed as previously described in ref. 18. Briefly, $2 \times 10^5$ HEK293T cells were transfected using Lipofectamine 3000 transfection reagent (Invitrogen) with 1 μg of all-in-one pX plasmid, expressing CoCas9 and sgRNA, and 10 pmol of dsODNs; scramble sgRNA was used as negative control. The day after transfection, cells were detached and selected with 1 μg/ml puromycin as described in ref. 19 Three days after transfection, cells were collected, and genomic DNA extracted using NucleoSpin Tissue Kit (Macherey-Nagel) following manufacturer's instructions. Using Covaris S200 sonicator, genomic DNA was sheared to an average length of 500 bp. End-repair reaction was performed using NEBNext Ultra End Repair/dA Tailing Module and adaptor ligation using NEBNext® Ultra™ Ligation Module, as described by ref. 56 Amplification steps were then performed following the GUIDE-seq original protocol[18].

Visualization of aligned off-target sites is available as a color-coded sequence grid (Supplementary Fig. 7a–d and Supplementary Fig. 9d–h). GUIDE-seq data are provided as Source data files.

### Western blot

Cell pellets from cells transfected with Cas9s expression plasmids and control sgRNAs were collected after 24, 48, and 72 h and lysed with RIPA buffer. Ten μg of the whole lysates quantified by BCA assay (Thermo Fisher Scientific) were run on a SDS-PAGE gel and transferred to a PVDF membrane. Membranes were probed with mouse anti-V5 antibodies (1:1000 dilution, Thermo Fisher Scientific, #46-0705, clone SV5-Pk1) recognizing the V5 tag fused to Cas9 and mouse anti-α-Tubulin antibodies (1:6000 dilution, Sigma Aldrich, #T6074, clone B-5-1-2). Detection was performed with Pierce ECL reagent (Thermo Fisher Scientific) using the Uvitec Alliance Instrument.

### Subcellular fractionation

HEK293T cells (500,000 cells/well) were seeded on a 6-well plate and transfected with 2.5 μg of Cas9s expression plasmids (pX-Cas9s) and 1.25 μg of pUC-sgRNA. Pellets were collected after 72 h and resuspended in Nuclear Resuspension Buffer (NSB) (10 mM HEPES pH 8, 10 mM KCl, 1.5 mM MgCl2, 0.34 M Sucrose, 10% Glycerol, 1 mM DTT,

0.1% TritonX, 1X Protease inhibitors), followed by cytoplasmic and nuclear fractions separation by sequential centrifugations. Samples' concentration was assessed through Bradford assay (Sigma-Aldrich) and 7.5 μg of each fraction were analyzed by Western blot. Cas9s were detected with anti-V5 antibodies (1:1000 dilution, Thermo Fisher Scientific, #46-0705, clone SV5-Pk1), using mouse anti-GAPDH (1:4000 dilution, Santa Cruz Biotechnology, #sc-32233, clone 6C5) and rabbit anti-H3 (1:10000 dilution, Abcam, #ab1791) as loading control and to verify the purity of the subcellular fractions. Goat anti-Mouse (1:15000, dilution, KPL, #0741809) or goat anti-Rabbit (1:10000 dilution, Santa Cruz Biotechnology, #sc-2004) HRP-conjugated were used as secondary antibodies.

### Statistics and reproducibility

A size of three or more was used for all experiments, as a minimum of three independent samples are required to perform statistical tests. No other methods were used to determine sample size; no data were excluded from the analyses; the experiments were not randomized; the Investigators were not blinded to allocation during experiments and outcome assessment. Statistical significance tests were performed using GraphPad Prism (version 9.4.1). Data in Fig. 4c, Supplementary Figs. 4 and 10b were analyzed with a one-way ANOVA followed by a two-sided Holm-Šídák test. Data in Fig. 4a and Supplementary Fig. 5a were analyzed using a two-way ANOVA corrected for multiple comparisons using the Holm-Šídák method, while a two-sided t-test was used in Fig. 5g and Supplementary Fig. 5b. For all analyses, adjusted p-values less than 0.05 were considered statistically significant.

### Reporting summary

Further information on research design is available in the Nature Portfolio Reporting Summary linked to this article.

## Data availability

Source data are provided with this paper. Sequencing data for PAM determination assays, GUIDE-Seq experiments and deep sequencing of mouse retinas are publicly available at NCBI Sequence Read Archive (PRJNA1088104) [https://www.ncbi.nlm.nih.gov/sra/PRJNA1088104]. Source data are provided with this paper.

## Code availability

Custom code used in this study is publicly available at https://github.com/Matteo-Ciciani/CoCas9-data-analysis[57].

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

## Acknowledgements

We are grateful to Cereseto's lab members for helpful discussion throughout the project. We thank the Next Generation Sequencing facility at the University of Trento for technical support. We thank Aitor Blanco-Miguez, Moreno Zolfo and Francesco Asnicar from Segata's lab for helping in developing the CRISPR search computational pipeline. This work was supported by the European Union's Horizon 2020 innovation programme through the UPGRADE (Unlocking Precision Gene Therapy) project (grant agreement No 825825) to ACe and AA and the Horizon Europe EIC Pathfinder programme AAVolution (grant agreement 101071041) to A.Ce. and A.A.; by the European Research Council (ERC-STG project MetaPG-716575) to N.S. and by the National Cancer Institute of the National Institutes of Health (1U01CA230551) to N.S.

## Author contributions

M.D., E.P., E.V., L.P, F.E., S.A. designed and performed the experiments; M.C. and N.S. developed the computational pipeline for CRISPR loci search; M.C., M.D., E.P., E.V., L.P., L.L., I.B., G.M., S.A., M.L., and M.C. collected and analyzed the data; A.Ce., N.S., A.A., A.M., A.Ca., F.E., E.P., M.D., E.V., L.P., M.C. conceived and designed the study, wrote and edited the paper; A.Ce. and N.S. were responsible for the coordination of the study. All authors read, corrected, and approved the final manuscript.

## Competing interests

The authors declare competing financial interests: A.Ce. is a co-founder and holds stocks of Alia Therapeutics, a genome editing company. A.Ca. is a co-founder, holds stocks and is currently an employee of Alia Therapeutics. L.P. is an employee of Alia Therapeutics. M.C. and N.S. are consultants of Alia Therapeutics. AA is co-founder and shareholder of InnovaVector. Pending patent applications (WO2023118349 and WO2024056880, claiming priority to USPTO applications) covering nucleases presented in this work have been filed by Alia Therapeutics. The remaining authors declare no competing interests.
