## [Peer Review File · Nature Communications]

CoCas9 is a compact nuclease from the human microbiome for efficient and precise genome editingEditorial Note: This manuscript has been previously reviewed at another journal that is not operating a transparent peer review scheme. This document only contains reviewer comments and rebuttal letters for versions considered at *Nature Communications*.

Reviewer #1 (Remarks to the Author):

Although not directly compared to other recently developed small cas variants, this study shows the potential of new small cas variants found in the microbiome. The revised manuscript, as a whole, is of sufficient quality to support that the variant is substantially acceptable in its natural state of efficiency and ease of use and worthy of publication in Nature Communications.

For the comment 3 by the reviewer 4, based on Fig3b (inconsistent with SupFig5a), on-target indel frequencies for HPRT and ZSCAN2 is too low for CoCas9, raising a concern if off-target comparison using these sites with SpCas9 is fair. With regard to Nme2Cas9, if the authors argue that a direct comparison between Nme2Cas9 and CoCas9 is not appropriate due to PAM incompatibilities, the high accuracy of the original Nme2Cas9 demonstrated in a similar way (Edraki et al 2019) should be mentioned.

For the comment 4 by the reviewer 4, the authors presumably performed lentiviral transduction and subsequent antibiotic selection (if so, please mention in both main text and figure legend) using three types of human primary cells (Fig 5a-c). Please also explain why statistical assessment is missing for Fig 5a.

This reviewer feels that the authors have adequately responded to the remaining comments made by reviewer 4.

Reviewer #1 (Remarks to the Author):

Although not directly compared to other recently developed small cas variants, this study shows the potential of new small cas variants found in the microbiome. The revised manuscript, as a whole, is of sufficient quality to support that the variant is substantially acceptable in its natural state of efficiency and ease of use and worthy of publication in Nature Communications.

We thank the reviewer for the positive assessment of our study.

For the comment 3 by the reviewer 4, based on Fig3b (inconsistent with SupFig5a)

Figure 3b and Supplementary Figure 5a report the editing of CoCas9 in the same loci but with different sgRNAs, since in Sup5a the goal was to use sgRNAs compatible with the PAM of SpCas9 for a fair comparison (not needed in 3b).

, on-target indel frequencies for HPRT and ZSCAN2 is too low for CoCas9, raising a concern if off-target comparison using these sites with SpCas9 is fair.

As follow up from the former point, the sgRNAs selected for the off-target analysis were not selected from Figure 3b, rather from Supplementary Figure 5a to ensure a fair off-target comparison as clearly stated in main text (Lines 132-136). We specifically chose sites where the on-target editing was similar between CoCas9 and SpCas9, as reported in Supplementary Figure 5a.

With regard to Nme2Cas9, if the authors argue that a direct comparison between Nme2Cas9 and CoCas9 is not appropriate due to PAM incompatibilities, the high accuracy of the original Nme2Cas9 demonstrated in a similar way (Edraki et al 2019) should be mentioned.

We have added a sentence in the discussion with this citation as suggested by the Reviewer.

For the comment 4 by the reviewer 4, the authors presumably performed lentiviral transduction and subsequent antibiotic selection (if so, please mention in both main text and figure legend) using three types of human primary cells (Fig 5a-c).

We thank the Reviewer for this suggestion and we have added the selection step in the manuscript both in the main text and in the legend of Figure 5.

Please also explain why statistical assessment is missing for Fig 5a.

The original version reported a single experiment, we have now repeated the experiment and reported data with statistical analysis.

This reviewer feels that the authors have adequately responded to the remaining comments made by reviewer 4.